# A Retrospective Chart Review of Treatment Patterns and Overall Survival among a Cohort of Patients with Relapsed/Refractory Mycosis Fungoides in France, Germany, Italy, Spain and the United Kingdom

**DOI:** 10.3390/cancers15235669

**Published:** 2023-11-30

**Authors:** Chalid Assaf, Timothy M. Illidge, Nathalie Waser, Mary He, Tina Li, Athanasios Zomas, Nawal Bent-Ennakhil, Meredith Little, Pablo L. Ortiz-Romero, Nicola Pimpinelli, Mehul Dalal, Martine Bagot

**Affiliations:** 1Department of Dermatology, Helios Klinikum Krefeld, Lutherplatz 40, 47805 Krefeld, Germany; 2Institute of Molecular Medicine, Medical School Hamburg, 20457 Hamburg, Germany; 3Manchester NIHR BRC Centre, Christie Hospital, University of Manchester, Manchester M20 4BX, UK; tim.illidge@manchester.ac.uk; 4ICON Plc, 3455 North Service Road, Unit 400, Burlington, ON L7N 3G2, Canada; nathalie.waser@iconplc.com (N.W.); maryhe0229@me.com (M.H.); tina.ljj@gmail.com (T.L.); 5Takeda Pharmaceuticals International AG, Thurgauerstrasse 130, CH-8152 Zurich, Switzerland; athanasios.zomas@takeda.com (A.Z.); nawal.bentennakhil@takeda.com (N.B.-E.); 6Takeda Development Center Americas, Inc. (TDCA), Lexington, MA 02421, USA; mehul.dalal@takeda.com; 7Department of Dermatology, Hospital 12 de Octubre, Institute I+12, CIBERONC, Medical School, Universidad Complutense, 28041 Madrid, Spain; pablo.ortiz@salud.madrid.org; 8Department of Health Sciences, Section Dermatology, Hospital P. Palagi, University of Florence Medical School, Viale Michelangiolo 41, 50125 Florence, Italy; nicola.pimpinelli@unifi.it; 9Saint-Louis Hospital, Université Paris Cité, 1 Avenue Claude Vellefaux, 75010 Paris, France; martine.bagot@aphp.fr

**Keywords:** mycosis fungoides, relapsed/refractory, treatment patterns, Europe, real-world data, observational study, overall survival, progression-free survival

## Abstract

**Simple Summary:**

Here, we report real-world treatment patterns and the overall survival of 104 patients with relapsing/refractory (R/R) MF who had received first-line therapy between 1984 and 2016. Our study found that second- and third-line therapies for R/RMF were heterogeneous, including systemic therapies, radiotherapy and topical therapies. More than two-thirds of patients at second- and third-line treatment received chemotherapy, and we observed that the median overall survival was longer with non-chemotherapy (not reached) vs. chemotherapy (6.5 years) treatment, suggesting that non-chemotherapy options may be more beneficial to patients.

**Abstract:**

(1) Background: Most patients with mycosis fungoides (MF), a form of cutaneous T-cell lymphoma (CTCL), develop relapsed/refractory (R/R) disease following front-line systemic therapy. This report describes treatment patterns and outcomes from the subpopulation with R/R MF. (2) Methods: This observational, retrospective, cohort study analyzed patient records (1984–2016) from 27 clinical sites in Europe. Outcomes included treatments received, response to first-, second- and third-line treatment, overall survival (OS) and progression-free survival (PFS). (3) Results: Of 104 patients with MF, 100 received second-line and 61 received third-line therapy. The median (range) times from the start of first-line therapy to the first R/R MF and from the first to the second R/R MF were 11.2 (0.3–166.5) and 13.5 (0.0–174.6) months, respectively. Second-and third-line treatment options varied and comprised systemic therapies (85% and 79% of patients, respectively), radiotherapy (32% and 34%, respectively) and topical therapies (48% and 36%, respectively). The median (95% confidence interval [CI]) OS from the diagnosis of the first R/R MF was 11.5 (6.5–not reached [NR]) years and was higher with non-chemotherapy (NR) versus chemotherapy (6.5 years); the estimated median PFS (95% CI) from the time of the first R/R MF was 1.3 (1.0–2.1) years. (4) Conclusions: High rates of R/R disease were observed after second- and third-line treatments in this real-world cohort, with longer median OS in patients receiving non-chemotherapy treatment versus chemotherapy. Following the standard management of MF and using recently approved targeted therapies can help improve patient outcomes in advanced-stage MF.

## 1. Introduction

Mycosis fungoides (MF) is the most common form of cutaneous T-cell lymphoma (CTCL) [1]. MF is a rare disease and more common in males than females (ratio 1.4–1.6) [2,3]. The prevalence of MF is higher in Europe than in North America [4] and the proportion of patients with MF in relation to the overall CTCL population varies between countries: e.g., Germany (62%) [5] and Italy (64%) [6] versus France (41%) [7], the United Kingdom [55%] [8] and Spain [59%] [9]. According to the International Society for Cutaneous Lymphomas (ISCL) and the European Organization for Research and Treatment of Cancer (EORTC; ISCL/EORTC), patients with MF are classified into early-stage (IA–IIA) and advanced-stage (IIB–IVB) [10]. Most patients are diagnosed with early disease at a median age of 58 years [11]. Five-year overall survival (OS) rates vary according to disease stage in patients with MF [3], ranging from 93.6% and 83.8% at stages IA and IB, to 60%, 45.9% and 44.9% at stages IIB, IIIB and IV, respectively [12]. Compared with patients at stage IA, the age- and sex-adjusted relative risks of death for patients at stages IB and IIA are 1.3 and 3.5, respectively [11].

Due to the heterogeneous and progressive nature of MF, treatment plans are multimodal and vary depending on clinical stage, subtype and location of involvement [13,14]. In Europe, the treatment of MF is usually based on the EORTC consensus recommendations [15,16]. However, achieving and maintaining a durable response remains challenging, particularly in patients with advanced-stage MF. Most patients who receive front-line therapy, which includes skin-directed (early-stage MF) and systemic treatments (e.g., chemotherapy in advanced-stage MF) alone or in combination, eventually relapse or develop treatment-refractory disease [14]. In such cases, several novel therapies are available to these patients, including monoclonal antibodies (mogamulizumab) [17], histone deacetylase inhibitors (vorinostat, belinostat, romidepsin) [18] and antibody–drug conjugates, such as brentuximab vedotin [19,20]. Nevertheless, among patients with relapsed or refractory (R/R) MF, a knowledge gap exists regarding patient characteristics, treatment patterns and outcomes with these novel treatments. This study aimed to assess current treatment patterns and outcomes among patients with R/R CTCL in a real-world setting. We previously reported the results from 157 patients with R/R CTCL after systemic therapy across five European countries, with a median follow-up time of 5.6 years from diagnosis [20]. Here, we report real-world treatment patterns and OS from the same study in the subgroup of patients with MF.

## 2. Materials and Methods

### 2.1. Study Design and Patients

The study design has been previously described [20]. Briefly, this was a retrospective, multicenter, chart review study involving 27 participating clinical European sites in France, Germany, Italy, Spain and the UK. The sites were carefully selected to include a representative sample considering size, type (academic versus community) and geography (spread across each country) to capture potential variability in treatment practices. Within the retrospective design, patient charts were based on dates of diagnosis of R/R disease, allowing all patients to be included regardless of their outcome to prevent immortal time bias. Given the rarity of patients with MF, no other sampling techniques to reduce bias were considered. Eligible patients were aged ≥18 years with a diagnosis of CTCL, including MF, with R/R disease after a first course of systemic therapy before 1 January 2016. The index date was defined as the date of diagnosis of R/R disease (1984–2016). Patients were excluded if they received systemic therapy before having R/R disease as part of a clinical trial. While the primary study included three lines of therapies [20], the current analysis focused on patients who received second- and third-line therapies. Data from patients’ medical charts were extracted from the study sites into an electronic data capture system (September 2017–November 2018) and included patient demographics and clinical characteristics (CTCL subtype and disease stage at index), treatments and responses to treatment. This study was conducted in accordance with the International Society for Pharmacoepidemiology (ISPE) guidelines for Good Pharmacoepidemiology Practice [21] and the Strengthening the Reporting of Observational Studies in Epidemiology (STROBE) guidelines [22]. Patient data were pseudonymized following both central and local ethics committee approvals in accordance with local regulations, as specified previously [20].

### 2.2. Outcome Measures

Outcomes included treatments received by line of therapy (systemic therapy, radiotherapy, skin therapy and other therapies [e.g., surgery, transplant]); response to treatment, both global and local (including the time from the start of a line of therapy to R/R disease); the stage at index date in patients who received second-line treatment; OS and progression-free survival (PFS). MF stages I–IV were determined according to the ISCL/EORTC classification [10]. Outcome was defined by treatment type e.g., non-chemotherapy versus chemotherapy. Chemotherapies included in this cohort were chlorambucil, gemcitabine, pegylated liposomal doxorubicin, CHOP (i.e., cyclophosphamide, doxorubicin, vincristine, prednisone), CHOP-like polychemotherapies, EPOCH and EPOCH-like multi-agent chemotherapy.

The global response assessment was based on tumor–node–metastasis–blood (TNMB) staging (i.e., skin, lymph nodes, viscera and blood); where a global response assessment was not available in patients’ charts, skin (local) response alone was recorded. Although safety was an outcome measure in the previously published full CTCL population [20], it was not specifically analyzed in the MF subpopulation.

### 2.3. Statistical Analyses

As the study was descriptive, no formal sample size calculation was performed. Death, to assess OS, and death or progression, to assess PFS, were estimated from the index date using Kaplan–Meier estimates with 95% confidence intervals (CI). Progression was defined as progression from early- to advanced-stage MF, developing R/R MF or initiation of subsequent therapy after treatment for the first R/R MF. All other analyses were summarized using descriptive statistics (mean and standard deviation [SD] or median and range [minimum–maximum] for continuous variables; numbers and percentages for categorical variables). Dates needed to have both the year and month included in the time interval calculation; if the day was missing from a date, 15 was imputed as the day.

## 3. Results

In total, 104 patients with MF who had received first-line therapy were included in this retrospective study. The median age was 54.5 years, and most patients were male (60.6%; Table 1).

The physician specialty by country for the overall population is presented in Table 2. The data show that the dermatologists are the primary physician caring for the MF patients. However, hematologists are often involved in the management of MF patients in Italy, Spain and UK followed by France and Germany.

### 3.1. Disease Stage and Treatment Pattern by Country at Diagnosis or First R/R Disease

The most common ISCL/EORTC stages reported at diagnosis or at the time of first R/R disease IB or IIB are shown in Table 3.

As first-line treatment prior to R/R1, all 104 patients received systemic therapy (see detailed distribution of systemic therapies in Table 4). Of these, most patients received interferon-alpha (IFN, 26%) followed by bexarotene (22%). Interestingly, 25% of all patients received single agent or combination therapy prior to R/R1.

There are also differences regarding the choice of systemic treatment prior to R/R1 between countries, as shown in Table 5. As mentioned before, interferon-alpha was the most commonly used systemic therapy, despite differing in a wide range regarding country distribution, from 9.5% in France to 41.2% in the UK. Regarding the distribution of the use of chemotherapy, the most common use as single or combination chemotherapy was found in Spain (44.4%), followed by (35%) and the UK (26.4%).

Of these 104 patients, 100 went on to receive second-line therapy, and 61 received third-line therapy (Figure 1).

Of 81 patients with an available global response assessment to first-line systemic therapy or radiotherapy, 7 (6.7%) had a complete response (CR) and 30 (28.8%) had a partial response (PR). Local response assessments were available for 13 patients; 1 had a CR (1.0%), while 8 (7.7%) had a PR (Table 6).

New lesions were present in 64 (64.0%) patients with first R/R disease. The median (range) time from diagnosis to first R/R disease was 11.2 (0.3–166.5) months. The median (range) time from diagnosis of the first R/R disease to the last known status was 3.5 (0.0–20.7) years.

### 3.2. Second-Line Treatment Patterns

Among 100 patients who received second-line therapy, 85 (85.0%) received systemic treatment, 32 (32.0%) radiotherapy, 48 (48.0%) skin-directed therapies and 10 (10.0%) other treatments. Of those receiving systemic therapy, 20 (23.5%) were treated with single-agent chemotherapy and 11 (12.9%) with combination chemotherapy. Radiotherapy and skin-directed therapy were mostly delivered via electron beam (*n* = 16 [50.0%]) and topical steroids (*n* = 35 [72.9%]), respectively (Table 7).

### 3.3. Response to Second-Line Treatment

Overall, 94 patients received second-line systemic therapy or radiotherapy. Of these, 69 had an available global response assessment; 12 (12.8%) had CRs and 31 (33.0%) had PRs. A local response assessment was available for 14 patients; 1 (1.1%) had a CR and 7 (7.4%) had PRs (Table 3). Of the total population (*n* = 100) with MF who received second-line therapy, 80 (80.0%) patients had a second R/R MF. The median (range) time from first to second R/R MF was 13.5 (0.0–174.6) months (Table 8).

In patients who received systemic therapy or radiotherapy as second-line treatments, 31/94 (33.0%) discontinued treatment, mainly due to disease progression (*n* = 7; 22.6%) and toxicity (*n* = 6; 19.4%; Table 9).

### 3.4. Third-Line Treatment Patterns

Of the 80 patients with R/R MF after second-line treatment, 61 (76.3%) received third-line treatment and 19 (23.8%) remained untreated during data collection. A total of 48 (78.7%) patients received systemic therapy, 21 (34.4%) radiotherapy and 22 (36.1%) skin-directed therapy. Among those receiving systemic therapy, 14 (29.2%) received single-agent chemotherapy and 6 (12.5%) combination chemotherapy. Radiotherapy and skin-directed therapy were mostly delivered via electron beam (*n* = 13, 61.9%) and topical steroids (*n* = 19, 86.4%), respectively (Table 4).

### 3.5. Response to Third-Line Treatment

Of the 61 patients who received a third-line therapy, 38 (62.3%) subsequently relapsed or became refractory to treatment. In total, 56 (91.8%) patients received third-line systemic therapy or radiotherapy. Of these, 39 had an available global response assessment; 7 (12.5%) had CRs and 15 (26.8%) had PRs. A local response assessment was available for 10 patients; 5 (8.9%) had PRs (Table 3). The median (range) time from second to third R/R MF was 9.1 (0.0–39.0) months (Table 5). Overall, 20/56 (35.7%) patients who received systemic or radiotherapy as a third-line treatment discontinued therapy (Table 6). Regarding the survival outcomes at the last follow-up (11th March 2019), 39 (37.5%) patients died at a mean (SD) age of 65.2 (11.6) years, primarily due to CTCL (*n* = 18, 46.2%) and CTCL-related complications or toxicity (*n* = 10, 25.6%); the remaining deaths were due to other causes (*n* = 7, 17.9%) or were unknown (*n* = 4, 10.3%). Of the 63 (60.6%) patients known to be alive at the time of analysis, 57 (90.5%) had unresolved CTCL. The median OS (95% CI) in patients with first relapsed/refractory MF was 11.5 years (6.5–not reached [NR]). Patients receiving non-chemotherapy regimens had prolonged survival (*n* = 54; NR [12.1–NR] years) versus patients treated with chemotherapy (*n* = 31; 6.5 [2.7–NR] years) (Figure 2).

The median PFS for all 104 patients from the time of the first R/R MF was 1.3 (95% CI 1.0–2.1) years (Figure 3).

## 4. Discussion

Existing clinical research in MF focuses on advanced-stage disease, with a paucity of data on R/R MF. As such, we conducted this observational, retrospective cohort study to examine real-world treatment patterns and outcomes of patients with R/R MF across 27 clinics in Europe. Our data show that following first-line systemic treatment, 48% of patients relapsed and 52% developed refractory disease, and despite the administration of chemotherapy as second-line treatment, 80% of patients progressed to second R/R MF and approximately 60% experienced a third relapse or became refractory within 1 year. Treatment options at second and third lines were heterogenous, including systemic therapies in the majority of patients, radiotherapy (almost a third in both lines) and topical therapies. Although the rates of overall response from first to third-line treatment were similar, the time to relapse or refractory disease was significantly shortened, indicating that time-to-next treatment (TTNT) or PFS is a more meaningful endpoint reflecting the duration of treatment including its efficacy and safety [23].

The high rates of relapse and refractory disease were consistent with our previous analyses in similar cohorts in patients with CTCL and MF [20,24] and with a retrospective analysis of patients with MF and Sézary syndrome receiving a median of three lines of systemic therapies [25]. In line with our current data, despite the use of chemotherapy (more than two-thirds of patients at second and third lines), patients progressed after treatment for their first and second R/R MF within approximately 1 year. This is supported by reports of the limited durability of single or multi-agent chemotherapy at early treatment lines in controlling MF (median time to subsequent therapy, 3.9 months) [25]. For early-stage MF, EORTC recommends skin-directed therapies (e.g., topical corticosteroids, ultraviolet irradiation and mechlorethamine) for first-line treatment and systemic therapies (e.g., retinoids, interferon α-2b and low-dose methotrexate), possibly in combination with skin-directed therapy, for second-line treatment. These systemic therapies, in addition to others such as chemotherapy (gemcitabine, pegylated liposomal doxorubicin and cyclophosphamide, hydroxydaunomycin, vincristine and prednisolone [CHOP]), alemtuzumab and allogeneic stem cell transplant, are recommended for the first-line treatment of advanced-stage MF; single-agent and combination chemotherapy and allogeneic stem cell transplant are used as second-line therapies [13,16].

In our study, combination chemotherapy was used prior to R/R1 as a first-line systemic therapy in the range from 9.5% to 41.2%. In the relapsed or refractory situation, it was used in approximately 13% of patients as second and third lines. This heterogeneity, particularly in R/R MF, was higher than expected according to the European guidelines [15,16]. In addition to its use against aggressive diseases, other factors such as the specialty of the primary treating physician, e.g., dermatologist versus hematologist/oncologist, may play a role, as evidenced in our study by the correlation between the percentage of treating hematologists and the use of chemotherapy. Combination chemotherapy has limited durability in MF disease control and is best followed by allogeneic hematopoietic cell transplantation [14]. Almost a quarter of patients with second R/R MF were untreated or had unknown third-line treatment at the time of data extraction. This suggests that patients either relapse but do not show an aggressive disease and remain untreated, or that there is a lack of systemic treatments with an acceptable benefit–risk ratio beyond second-line treatment. Based on recent clinical trial data, such as the MAVORIC and ALCANZA post hoc analysis studies, using targeted therapies resulted in beneficial outcomes and a manageable safety profile in patients with advanced or R/R MF [19,26,27,28,29,30]. In addition, compared with other CTCL subtypes, patients with MF have been reported to have a longer OS from the time of the first R/R MF [20].

Here, patients with MF who received non-chemotherapy treatments had a longer OS than patients who received chemotherapy, suggesting that non-chemotherapy options may be more beneficial to patients. Our results indicate that the clinical burden of CTCL was likely to be considerable in the five European countries at the time of data collection. However, it is notable that this study spanned a period when using novel targeted therapies, such as mogamulizumab or brentuximab vedotin, was not common clinical practice in the R/R setting (1984–2016). Indeed, both targeted agents, the antibody–drug conjugate directing the CD30 molecule brentuximab vedotin and the monoclonal antibody against the CCR4 mogamulizumab, were approved in 2018 for patients who failed one prior systemic therapy. Recent data on both drugs have been shown to improve outcomes in this patient population [19,26,28,29,30,31]. For example, in the ALCANZA trial, patients with MF had global response rates of 50% with brentuximab vedotin versus 10% with methotrexate or bexarotene [30]; in the MAVORIC trial, treatment with mogamulizumab resulted in a superior PFS of 7.7 months versus 3.1 months with vorinostat in patients with R/R MF or Sézary syndrome [26]. The number of patients in our study having allogeneic stem cell transplantation (alloSCT) is small. This may be reflected by its limited role as it was used in the study period most often as a salvage therapy. To date, the body of evidence is limited to retrospective single center studies, registry data and small prospective studies with no randomized controlled trials. However, just recently, the first and hitherto only prospective trial on alloSCT in CTCL—the “CUTALLO trial”—was published, showing a significantly longer progression-free survival in advanced CTCL patients compared to patients with non-alloSCT [31]. These results may have, in the future, a changing effect in the treatment of the advanced CTCL patients in clinical practice.

Other targeted agents with novel mechanisms of action, such as checkpoint inhibitors and JAK inhibitors, are currently being explored in ongoing clinical trials [32]. Additional observational studies including targeted therapies are warranted to inform on the real-world impact of these novel treatments.

Also contributing to the heterogeneity of treatment approaches at the different disease stages was that patients were managed by multiple physicians with diverse specialties’, including dermatologists, oncologists and hematologists, as well as a variety of reference guidelines (e.g., EORTC, the European Society for Medical Oncology [ESMO] [33] and the National Institute for Health and Care Excellence [NICE]) [34]. Limitations are related to the retrospective nature of the study. Since the managing physicians collected the data for purposes other than to address the objectives of this study, specific clinical characteristics, treatments and outcomes might not have been standardized, as is the case in clinical trials. The level of detail of clinical records might vary depending on the chart recording practices of each study site. The small number of patients receiving second- and third-line therapies prevented the performance of sub-analyses on some outcomes (e.g., stratifying OS by types of chemotherapy or disease stage).

## 5. Conclusions

This observational study demonstrated a high rate of R/R MF following second- and third-line treatments in real-world settings in Europe, with a longer OS in patients receiving non-chemotherapy versus chemotherapy. This suggests that the clinical burden of R/R MF is significant in Europe. Adhering to standard management of MF and using recently approved targeted therapies may improve patient outcomes, including prolonging OS in advanced-stage MF.

## Figures and Tables

**Figure 1 cancers-15-05669-f001:**
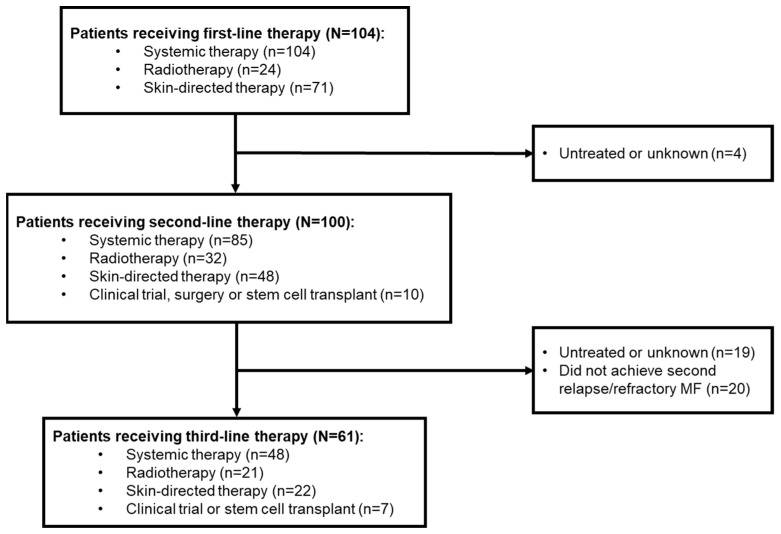
Patient disposition according to line of therapy. Numbers of patients receiving the different types of therapy add up to more than the total as treatments were received alone and in combination. MF, mycosis fungoides.

**Figure 2 cancers-15-05669-f002:**
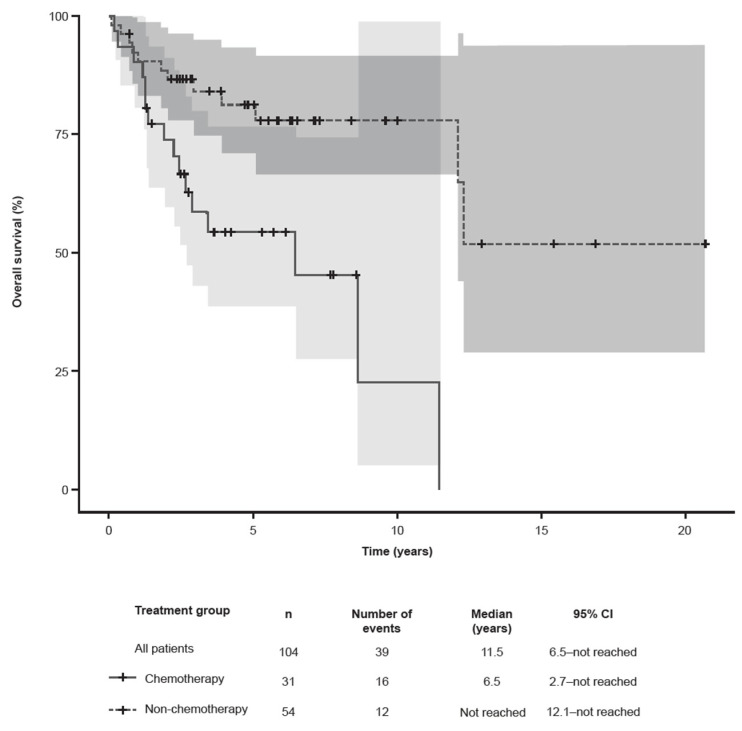
Kaplan–Meier estimates of overall survival in patients with R/R MF from the time of start of second-line therapy. The gray shades represent the 95% confidence intervals. CI, confidence interval; MF, mycosis fungoides.

**Figure 3 cancers-15-05669-f003:**
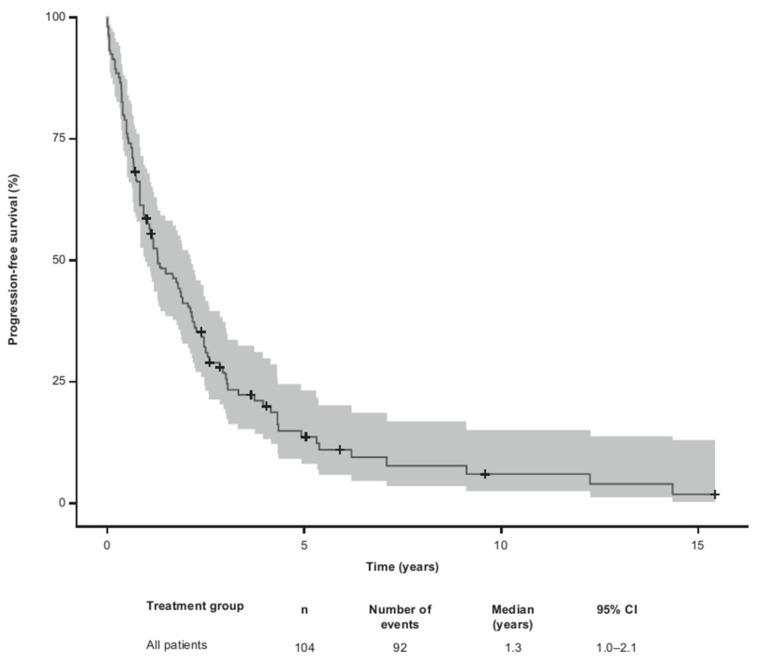
Kaplan–Meier estimates of progression-free survival from time of first relapsed/refractory MF. Defined as patients who progressed, relapsed or became refractory after therapy for first relapsed/refractory MF or deceased thereafter. The gray shade represents the 95% confidence intervals. CI, confidence interval; MF, mycosis fungoides.

**Table 1 cancers-15-05669-t001:** Demographics and characteristics of patients with MF at diagnosis.

Age	*n* = 104
Mean (SD)	54.3 (14.7)
Median (range)	54.5 (21.0–82.0)
Sex, *n* (%)	
Male	63 (60.6)
Female	41 (39.4)
Ethnicity, *n* (%)	
Caucasian	82 (78.8)
African	1 (1.0)
Asian	0 (0.0)
Caribbean	0 (0.0)
Latino	0 (0.0)
Multiracial	0 (0.0)
Other	1 (1.0)
Unknown	20 (19.2)
Specialty of physicians managing patients *, *n* (%)	
Family physician	22 (21.2)
Dermatologist	88 (84.6)
Oncologist	26 (25.0)
Dermatologist/oncologist	25 (24.0)
Hematologist	31 (29.8)
Hematologist/oncologist	12 (11.5)
Radiation oncologist	28 (26.9)
Other	6 (5.8)
Care setting, *n* (%)	
Community hospital	15 (14.4)
Academic teaching hospital	71 (68.3)
Cancer center	16 (15.4)
Community clinic	1 (1.0)
Other	0 (0.0)
Unknown	1 (1.0)

* Percentages may add up to more than 100 as multiple selections were allowed. MF, mycosis fungoides; SD, standard deviation.

**Table 2 cancers-15-05669-t002:** Specialty of diagnosing and treating physician of study subjects at diagnosis—by country.

	France	Germany	Italy	Spain	UK
	(%)	(%)	(%)	(%)	(%)
Family physician	15.8	38.1	7.5	20.5	28.9
Dermatologist	89.5	100.0	55.0	87.2	94.7
Oncologist	5.3	23.8	2.5	10.3	73.7
Dermatologist/oncologist	42.1	42.9	5.0	28.2	23.7
Hematologist	10.5	0.0	62.5	23.1	47.4
Hematologist/oncologist	10.5	4.8	20.0	20.5	13.2
Radiation oncologist	31.6	28.6	0.0	30.8	39.5
Other	0.0	4.8	2.5	10.3	7.9
Unknown	0.0	0.0	0.0	0.0	0.0

**Table 3 cancers-15-05669-t003:** Disease stage at diagnosis and at time of first relapsed or refractory MF.

ISCL/EORTC Staging, *n* (%)	At Diagnosis (*n* = 104)	At First R/R MF (*n* = 100)
IA	12 (11.5)	5 (5.0)
IB	42 (40.4)	26 (26.0)
IIA	4 (3.8)	4 (4.0)
IIB	19 (18.3)	28 (28.0)
IIIA	2 (1.9)	10 (10.0)
IIIB	4 (3.8)	3 (3.0)
IVA1	1 (1.0)	1 (1.0)
IVA2	4 (3.8)	6 (6.0)
IVB	2 (1.9)	2 (2.0)
Unknown	14 (13.5)	15 (15.0)

ISCL/EORTC, International Society for Cutaneous Lymphomas/European Organization for Research and Treatment of Cancer; R/R, relapsed or refractory; MF, mycosis fungoides.

**Table 4 cancers-15-05669-t004:** Details of systemic therapy prior to R/R1.

	MF (n = 104)	
	n	%
Systemic therapy	104	100.0
Bexarotene	22	21.2
METX	12	11.5
Bexarotene + METX	1	1.0
Other retinoids	13	12.5
Single-agent chemotherapy	15	14.4
Combination chemotherapy	10	9.6
Corticosteroids	3	2.9
IFN	27	26.0
HDAC inhibitors	0	0.0
ECP	1	1.0

**Table 5 cancers-15-05669-t005:** Treatment description at first systemic therapy/RT prior to R/R1 by country.

Variable	France	Germany	Italy	Spain	UK
	(%)	(%)	(%)	(%)	(%)
Systemic therapy					
Bexarotene	6.2	14.3	17.5	36.1	2.9
METX	37.5	33.3	5.0	5.6	8.8
Bexarotene + METX	6.2	0.0	0.0	0.0	2.9
Other retinoids	0.0	23.8	5.0	5.6	17.6
Single-agent chemotherapy	12.5	14.3	15.0	19.4	17.6
Combination chemotherapy	12.5	4.8	20.0	25.0	8.8
Corticosteroids	0.0	0.0	7.5	0.0	0.0
IFN	12.5	9.5	25.0	8.3	41.2
HDAC inhibitors	0.0	0.0	0.0	0.0	0.0
ECP	12.5	0.0	5.0	0.0	0.0

Abbreviations: R/R = relapse/refractory; METX = methotrexate; IFN = interferon; HDAC = histone deacetylase; ECP = extracorporeal photopheresis; Percentages may add up to more than 100 as multiple selections are allowed.

**Table 6 cancers-15-05669-t006:** Response to systemic therapy or radiotherapy in first, second and third lines.

*n* (%)	First Line (*n* = 104)	Second Line (*n* = 100)	Third Line (*n* = 61)
Patients with Systemic Therapy or Radiotherapy	104 (100.0)	94 (94.0)	56 (91.8)
Global response			
Complete	7 (6.7)	12 (12.8)	7 (12.5)
Partial	30 (28.8)	31 (33.0)	15 (26.8)
Stable	16 (15.4)	11 (11.7)	7 (12.5)
Progression	28 (26.9)	15 (16.0)	10 (17.9)
Unknown	23 (22.1)	25 (26.6)	17 (30.4)
Local response *			
Complete	1 (1.0)	1 (1.1)	0 (0.0)
Partial	8 (7.7)	7 (7.4)	5 (8.9)
Stable	3 (2.9)	2 (2.1)	3 (5.4)
Progression	1 (1.0)	4 (4.3)	2 (3.6)
Unknown	10 (9.6)	11 (11.7)	7 (12.5)

* Data from local responses were collected if global response was unknown.

**Table 7 cancers-15-05669-t007:** Types of treatment at second and third lines.

Treatment	Second Line (*n* = 100)	Third Line (*n* = 61)
	*n* (%)	*n* (%)
Systemic therapy	85 (85.0)	48 (78.7)
Bexarotene	20 (23.5)	9 (18.8)
Methotrexate	10 (11.8)	0 (0.0)
Bexarotene + methotrexate	2 (2.4)	2 (4.2)
Other retinoids	6 (7.1)	4 (8.3)
Single-agent chemotherapy	20 (23.5)	14 (29.2)
Combination chemotherapy	11 (12.9)	6 (12.5)
Corticosteroids	1 (1.2)	0 (0.0)
Interferon	12 (14.1)	10 (20.8)
Histone deacetylase inhibitors	2 (2.4)	2 (4.2)
Extracorporeal photopheresis	1 (1.2)	1 (2.1)
Radiotherapy	32 (32.0)	21 (34.4)
Type of radiotherapy *		
Electron beam	16 (50.0)	13 (61.9)
X-ray	11 (34.4)	8 (38.1)
Unknown	5 (15.6)	0 (0.0)
Extent of radiotherapy		
Local	21 (65.6)	13 (61.9)
Total	8 (25.0)	8 (38.1)
Unknown	3 (9.4)	0 (0.0)
Target location *		
Head	5 (15.6)	2 (9.5)
Face	5 (15.6)	3 (14.3)
Anterior trunk	8 (25.0)	6 (28.6)
Posterior trunk	8 (25.0)	2 (9.5)
Arms	7 (21.9)	3 (14.3)
Legs	3 (9.4)	2 (9.5)
Palms	7 (21.9)	6 (28.6)
Unknown	15 (46.9)	11 (52.4)
Skin-directed therapy *	48 (48.0)	22 (36.1)
Topical steroids	35 (72.9)	19 (86.4)
Topical mechlorethamine	1 (2.1)	1 (4.5)
Topical carmustine	3 (6.2)	1 (4.5)
Topical bexarotene	0 (0.0)	0 (0.0)
Tazarotene	0 (0.0)	0 (0.0)
Imiquimod	0 (0.0)	0 (0.0)
Heliotherapy	0 (0.0)	0 (0.0)
Photodynamic therapy	0 (0.0)	1 (4.5)
UVB phototherapy	1 (2.1)	1 (4.5)
nbUVB phototherapy	5 (10.4)	1 (4.5)
PUVA phototherapy	19 (39.6)	5 (22.7)
Other	1 (2.1)	1 (4.5)
Other treatments	10 (10.0)	7 (11.5)
Clinical trial	6 (6.0)	4 (6.6)
Surgery	2 (2.0)	0 (0.0)
Stem cell transplant	2 (2.0)	3 (4.9)
Allogeneic stem cell transplant	2 (2.0)	2 (3.3)
Autologous stem cell transplant	0 (0.0)	1 (1.6)
Unknown	0 (0.0)	0 (0.0)

* Percentages may add up to more than 100 as multiple selections were allowed. nbUVB, narrow-band ultraviolet B; PUVA, psoralen and ultraviolet A; UVB, ultraviolet B.

**Table 8 cancers-15-05669-t008:** Time to next relapsed or refractory disease from initiation of systemic therapy at first, second and third lines.

	First Line (*n* = 104) *	Second Line (*n* = 100) **	Third Line (*n* = 61) **
Patients with relapsed or refractory MF, *n* (%)			
Relapsed	50 (48.1)	44 (44.0)	15 (24.6)
Refractory	54 (51.9)	36 (36.0)	23 (37.7)
Neither	0	20 (20.0)	23 (37.7)
Time to relapsed or refractory MF, months			
Mean (SD)	21.3 (27.7)	24.6 (30.8)	10.4 (8.2)
Median (range)	11.2 (0.3–166.5)	13.5 (0.0–174.6)	9.1 (0.0–39.0)
Unknown, *n* (%)	8 (7.7)	2 (2.5)	0 (0.0)

* Date of first R/R disease minus date of start of earlier systemic therapy and radiotherapy, plus one to include last day, and divided by 30. ** Date of next R/R disease minus date of previous R/R disease, plus one to include last day, and divided by 30. Patients were required to have at least one non-missing year and one month to be included in the initial calculation of time interval; any missing days were imputed as 15 days. MF, mycosis fungoides; R/R, relapsed or refractory; SD, standard deviation.

**Table 9 cancers-15-05669-t009:** Discontinuations in patients receiving systemic therapy and radiotherapy.

	After First Line (*n* = 94)	After Second Line (*n* = 56)
	*n* (%)	*n* (%)
Number of patients who discontinued treatment	31 (33.0)	20 (35.7)
Reason for early discontinuation		
Toxicity	6 (19.4)	3 (15.0)
Compliance	0 (0.0)	0 (0.0)
Death	3 (9.7)	4 (20.0)
Disease progression	7 (22.6)	4 (20.0)
Patient decision	5 (16.1)	4 (20.0)
Other	9 * (29.0)	4 ** (20.0)
Not provided in chart	1 (3.2)	1 (5.0)
Number of patients who did not take the treatment as prescribed	6 (6.4)	1 (1.8)
Reason for not being taken as prescribed		
Toxicity	1 (16.7)	1 (100.0)
Missed dose (scheduled)	0 (0.0)	0 (0.0)
Missed dose (unscheduled)	3 (50.0)	0 (0.0)
Other	1 (16.7)	0 (0.0)
Not available in the chart	1 (16.7)	0 (0.0)

* Other reasons include difficulty accessing vein with a peripherally inserted central catheter line and an infection; difficulty with venous access and started extracorporeal photopheresis; high blood sugar; myocardial infarction; patient did not attend clinic for 13 months; pruritus and erythema; sepsis; treatment did not have an effect on disease; insufficient response to Targretin; tongue cancer, followed by surgery (neck dissection) and radiotherapy. ** Other reasons include extracorporeal photopheresis being stopped due to difficulty administering therapy and interferon stopped because of no improvement in skin; interferon was a possible cause of abnormal liver function test results; lack of effect on disease; patient did not respond to treatment; pruritus, erythema and stress (father was ill).

## Data Availability

The data presented in this study are available on request from the corresponding author.

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
