# Peer review of "A Retrospective Chart Review of Treatment Patterns and Overall Survival among a Cohort of Patients with Relapsed/Refractory Mycosis Fungoides in France, Germany, Italy, Spain and the United Kingdom"

_cancers, 2023, doi:10.3390/cancers15235669_

Round 1

Reviewer 1 Report

Comments and Suggestions for Authors

It is surprising to include patients with stage Ia in R/R MF (5 cases) I am not sure it is relevant (and event Ib with 26 cases) for R/R MF. Similarly the patients with advanced stages other than IIb ( only 10 IIIA,, 3 IIIb, 1 IVA1, 6 IVA2 and 2 IVB represent a very limited group. These proportions do not reflect the real-life issues for R/R MF . This may partly explain the very low number of allogenic SCT (only 2) in a series of 104 R/R MF seen in expert centres.

The number of patients included (104 for 27 centres from 5 countries) is limited.

Moreover it is difficult to compare second or third lines and outcomes, in patients with Ia,Ia vs IIb, II, IV stages. This is a major and limiting  factor in interpreting the data, which it would have been more interesting to present in sub-groups according to stage.

How do you explain the fact that no patients appear to have been treated with targeted therapy? Some of them were available in the study period. But again you have included a limited proportion of advanced stages.

As all patients, whatever their stage, are mixed together, the effect of chemotherapy versus no chemotherapy cannot be interpreted. This is a major bias because it is highly likely that those treated with chemotherapy were in the most advanced stages. 

Therefore, the conclusions of this work are limited by these major biase. Apart from observing that practices are heterogeneous, it is absolutely impossible to conclude either on the strategy or on the response to treatment.

 -a few typos to be corrected : page 1, line 38 a "e" is missing for Europe; page 4 a space to be deleted in "managing"

Author Response

Dear Editor,

We thank you and the reviewers for your comments regarding the initial version of our manuscript and appreciate the reviewer´s valuable suggestions. A point-by-point response to all comments is included below and modifications are highlighted in the revised manuscript.

We thank the reviewer for his comments regarding the initial version of our manuscript and appreciate the reviewer´s valuable suggestions. A point-by-point response to all comments is included below and modifications are highlighted in the revised manuscript.

Revier #1

  • It is surprising to include patients with stage Ia in R/R MF (5 cases) I am not sure it is relevant (and event Ib with 26 cases) for R/R MF. Similarly the patients with advanced stages other than IIb ( only 10 IIIA,, 3 IIIb, 1 IVA1, 6 IVA2 and 2 IVB represent a very limited group. These proportions do not reflect the real-life issues for R/R MF . This may partly explain the very low number of allogenic SCT (only 2) in a series of 104 R/R MF seen in expert centres.

Aim of the study was to describe the real life situation of patients who relapsed or who were refractory to first line systemic therapy. The study was not limited to only advanced stage paients. Since relapse occur also in early stage of the patients, these were included in this study.

The small number of patients having SCT is also reflected by its limited role as most often used as a late salvage therapy. To date, the body of evidence is limited to retrospective single centre studies, registry data and small prospective studies with no randomised controlled trials. at the study period. Just recently, the first and hitherto only trial on SCT in CTCL was published (de Masson et al. Lancet 2023) We have extended this discussion regarding the role on SCT in the revised manuscript.

  • The number of patients included (104 for 27 centres from 5 countries) is limited.

No sample size calculation was performed due to the descriptive nature of this study. A sample size of 200 i.e. 40 per country) was initially targeted including different rypes of CTCL. Here, in this analsyses rare subtypes of CTCL were excluded, leading to the 104 cases with mycosis fungoides.

  • Moreover it is difficult to compare second or third lines and outcomes, in patients with Ia,Ia vs IIb, II, IV stages. This is a major and limiting  factor in interpreting the data, which it would have been more interesting to present in sub-groups according to stage.

Indeed, a direct delineation of patient stage to treatment was not possible due to the heterogeneity of treatment and limited number of patients, which would lead to statistical non-significant small subgroups once seperated to each clinical stages.

  • How do you explain the fact that no patients appear to have been treated with targeted therapy? Some of them were available in the study period. But again you have included a limited proportion of advanced stages.

As described in the manuscript the study period was limited to patients treated in the period from 1984-2016. The two targeted therapies approved for CTCL are the antibody drug conjugate directing the CD30 molecule brentuximab vedotin and the monoclonal antibody against the CCR4 mogamulizumab. Brentuximab vedotin was approved in Europe by the EMEA in , and Mogamulizumab in , and available in Europe in 6/20. So both drugs were approved after the inclusion period of our patients. We have included these details in the revised manuscript.

  • As all patients, whatever their stage, are mixed together, the effect of chemotherapy versus no chemotherapy cannot be interpreted. This is a major bias because it is highly likely that those treated with chemotherapy were in the most advanced stages. 
  • Therefore, the conclusions of this work are limited by these major biase. Apart from observing that practices are heterogeneous, it is absolutely impossible to conclude either on the strategy or on the response to treatment.

Indeed, a direct delineation of patient stage to treatment was not possible due to the heterogeneity of treatment and limited number of patients, which would lead to statistical non-significant small subgroups once seperated to each clinical stages. Interestingly, taken all the data together, our results show that chemotherapy was often applicated including EPOCH- and EPOCH-like regimen, which are outside of the European and other national CTCL guidelines. In addition, our data show an reduced overall survival of patients receiving chemotherapy compared to the non-chemotherapy. This is in line to the only prospective trial in this topic published by Kaye et al. in the NEJM 1989, confirming our conclusion. We have extended this discussion including the limitations in the revised manuscript.

  • -a few typos to be corrected : page 1, line 38 a "e" is missing for Europe; page 4 a space to be deleted in "managing"

The typos were corrected. Thank you-

Reviewer 2 Report

Comments and Suggestions for Authors

The manuscript reported the results of retrospective chart review of treatment patterns and overall survival among a cohort of patients with relapsed/refractory mycosis fungoides in Europe. The authors concluded that non-chemotherapy options might be more beneficial to patients because median overall survival was longer with non-chemotherapy than with chemotherapy treatment. I have some concerns about the conclusion.

1.  Please clearly define "chemotherapy." Does it include bexarotene, methotrexate, interferon, or histone deacetylase inhibitors?

2. I am afraid that there should be selection bias. Patients who received chemotherapy may have had higher tumor burden than those who were treated with non-chemotherapy. Please describe disease stages at diagnosis and at first R/R MF in chemotherapy and non-chemotherapy group.

3. It is surprising that response to systemic therapy or radiotherapy in first, second, and third line are almost the same. Generally speaking, the response in the second and third lines is poorer than that in the first line. Please discuss it.

Author Response

Revier #2

The manuscript reported the results of retrospective chart review of treatment patterns and overall survival among a cohort of patients with relapsed/refractory mycosis fungoides in Europe. The authors concluded that non-chemotherapy options might be more beneficial to patients because median overall survival was longer with non-chemotherapy than with chemotherapy treatment. I have some concerns about the conclusion.

  1. Please clearly define "chemotherapy." Does it include bexarotene, methotrexate, interferon, or histone deacetylase inhibitors?

The definition of used chemotherapy is now described in detail in the introduction of the revised version.

  1. I am afraid that there should be selection bias. Patients who received chemotherapy may have had higher tumor burden than those who were treated with non-chemotherapy. Please describe disease stages at diagnosis and at first R/R MF in chemotherapy and non-chemotherapy group.

Indeed, we cannot exclude a bias due to small subgroups leading to statistical non-significant results, which is limitation of this study. However, we could show in the new additional table that a significant percentage of the patient received already as a first line treatment mono- or rather combination chemotherapy (new Tables: prior R/R1). We believe that in addition to the severity of the disease the specialty of the diagnosing and treating physician could have an influence in the choice of treatment. In our study we could show that the percentage of treating hematologists/oncologists are e.g. in Spain and Italy much higher compared to Germany and UK (new Table). Interestingly in these countries the use mono- or combination chemotherapy used was significantly higher (new Tab), indicating that hematologist/oncologists are using chemotherapy in earlier lines as potentially dermatologists are do. We included this valuable information in the revised version

  • It is surprising that response to systemic therapy or radiotherapy in first, second, and third line are almost the same. Generally speaking, the response in the second and third lines is poorer than that in the first line. Please discuss it.

Although the rates of overall response from 1. line to 3.line treatment were similar the time to relapse or refractory disease was significantly shortened indicating that time-to-next treatment (TTNT) or PFS is a more meaningful endpoint reflecting the duration of treatment including its efficacy and safety. We included this valuable information in the revised version.

Round 2

Reviewer 1 Report

Comments and Suggestions for Authors

Thank you to the authors for responding to the comments and improving their manuscript by also specifying the limitations of the study. No further comments. 

Reviewer 2 Report

Comments and Suggestions for Authors

I have no further comments.